# Demographic, socioeconomic and regional disparities in the coverage of water, sanitation and hygiene facilities in four South Asian Countries

**Md. Hasibul Islam Jitu⊕\*, Mohammad Shahed Masud⊕**

Institute of Statistical Research and Training, University of Dhaka, Dhaka, Bangladesh

\* mhjitu@isrt.ac.bd

## Abstract

### Background

Ensuring an adequate water, sanitation, and hygiene (WASH) is crucial for upholding public health and achieving Sustainable Development Goals (SDG-6). The main goal of this study was to review the existing WASH facilities, mapping for regional comparisons, and identify the significant socioeconomic and demographic factors associated with WASH facilities in Afghanistan, Bangladesh, Nepal, and Pakistan.

### Methods

This study employed a quantitative research design using the most recent Multiple Indicator Cluster Survey (MICS) data from Afghanistan (2022–23, n = 23,213), Bangladesh (2019, n = 61,242), Nepal (2019, n = 12,655), and Pakistan (2017–19, n = 96,105). Data analysis was done using descriptive statistics and multivariate logistic regression model. Besides, spatial mapping was used for regional comparison, the Generalized Variance Inflation Factor (GVIF) was applied for checking multicollinearity, and the Receiver Operating Characteristic (ROC) curve was used to evaluate model performance.

### Results

This study revealed substantial disparities in the coverage of WASH facilities across four countries. Nepal had the highest coverage (75.33%), followed by Pakistan (59.47%), Bangladesh (50.28%) and Afghanistan (33.54%). Wealthier households were associated with higher odds of WASH facilities compared to the poor: Afghanistan (aOR = 7.83; 95% CI: 6.58–9.32; $p < 0.001$), Bangladesh (aOR = 5.75; 95% CI: 5.34–6.20; $p < 0.001$), Nepal (aOR = 5.80; 95% CI: 4.52–7.44; $p < 0.001$), and Pakistan (aOR = 9.64; 95% CI: 8.79-10.58; $p < 0.001$). In addition, place of residence, education of household head, access to the media, age of household head, and family size emerged as significant determinants of WASH facilities across all four countries.

**Data availability statement:** Datasets are publicly available at: https://mics.unicef.org/surveys

**Funding:** The author(s) received no specific funding for this work.

**Competing interests:** The authors have declared that no competing interests exist.

## Conclusion

The findings suggest that wealthier households, those with educated heads, and those in urban areas have higher coverage of WASH facilities. To ensure adequate WASH facilities, policymakers should focus on rural areas, lower-income groups, less educated household heads, and should conduct awareness campaigns.

## Introduction

Availability of adequate water, sanitation, and hygiene (WASH) is a fundamental human right and essential for public health and promoting overall development and growth [1,2]. WASH facilities are crucial for preventing waterborne illnesses like diarrhea, reducing child mortality, and enhancing overall well-being [3–5]. An estimated 88% of diarrheal disease, as noted by Ahmed *et al.* (2021) [6] and 1.4 million of deaths worldwide could be prevented by adopting proper WASH practices, according to Gordon *et al.* (2023) [5]. Sustainable Development Goal 6 (SDG 6) aims to ensure that everyone has access to reliable water and sanitation services by 2030, acknowledging their critical importance for sustainable development and public health [7]. However, WHO has raised concerns about insufficient global progress towards achieving access to safe WASH coverage by 2030. Besides, it warned that without significant acceleration, billions will lack access to these essential services [4,8].

The lack of adequate WASH facilities constitutes a critical global health challenge with far-reaching implications. Inadequate WASH access is a significant contributor to diarrheal diseases, the COVID-19 pandemic, and the spread of numerous Neglected Tropical Diseases (NTDs) including trachoma, guinea worm, schistosomiasis, buruli ulcer, and soil-transmitted helminths [2,4]. These health issues are particularly severe in low- and middle-income countries (LMICs) where infrastructure development is often insufficient [9]. The consequences of poor WASH facilities extend beyond immediate health risks, leading to higher child mortality and morbidity rates, severe child nutritional deficiency, poor educational outcomes due to inadequate school hygiene, and increased vulnerability during natural disasters [10–12].

Despite advancements in the past decade, nearly half the worldwide population lacks sufficient access to proper sanitation and safe drinking water [5]. Globally, nearly 2 billion peoples are without access to clean drinking water, 3.6 billion don't have adequate sanitation, and 2.3 billion are without basic hygiene services [4]. It's significantly impacting public health, resulting in 74 million life years lost to disability and over 1.4 million preventable deaths [5]. Poor WASH access leads to about 1.25 million deaths annually in LMICs, including 525,000 under five children. WHO has estimated that 5 million under-five deaths in 2020 could be prevented through safe WASH practices [13]. In South Asia, particularly in Afghanistan, Bangladesh, Nepal, and Pakistan, significant advancements have been made in enhancing access to water and sanitation services. However, over 134 million people still don't have access to safe drinking water in this region, with contamination levels in water sources ranging from 68 to 84 percent [14].

Research highlights various factors influencing WASH practices, including geographic location, economic barriers, political commitment, household conditions, household income, ethnicity, and education [15–17]. Although various studies have examined WASH facilities among specific segments of the population in Afghanistan [18,19], Bangladesh [20,21], Nepal [22], and Pakistan [23,24], they do not comprehensively capture households access to WASH facilities and the associated factors at a national level, with the exception of one previous country level study in Bangladesh [6]. This shows that households access to WASH facilities

and the associated factors in Afghanistan, Bangladesh, Nepal, and Pakistan are unexplored. To the best of our knowledge, no studies have been conducted that compare the coverage of WASH facilities and identify the significant factors influencing these facilities in this region. The lack of cross-country comparisons within the region hampers a comprehensive assessment of factors influencing WASH coverage.

The major objective of this research is to review the existing WASH facilities and identify the significant socioeconomic and demographic factors associated with WASH in this region. We are also interested in country-wise and region-wise comparisons of household-level coverage of WASH facilities and understanding the distribution of these facilities across the studied areas by socioeconomic and demographic factors.

Studying access to WASH facilities is crucial for monitoring progress towards achieving SDGs. By examining the factors influencing accessibility and utilization of WASH facilities, the study seeks to fill gaps in existing knowledge and provide targeted strategies for improving WASH facilities in this region.

## Methods

### Study countries and data sources

The MICS data collecting from the corresponding four south Asian countries: Afghanistan, Bangladesh, Nepal, and Pakistan, serve as data source for this study [38]. When multiple datasets were available for the same country within a specific time frame, the most recent survey was selected (Fig 1). The MICS surveys employed a two-stage sampling strategy. Primary Sampling Units (PSUs) or enumeration area were systematically selected within each stratum in the initial stage. In the second stage, a specific number of Secondary Sampling Units (SSUs) or households, were sampled within each selected PSU. This method ensured a representative sample across urban and rural areas in each country. Each of these survey protocol received

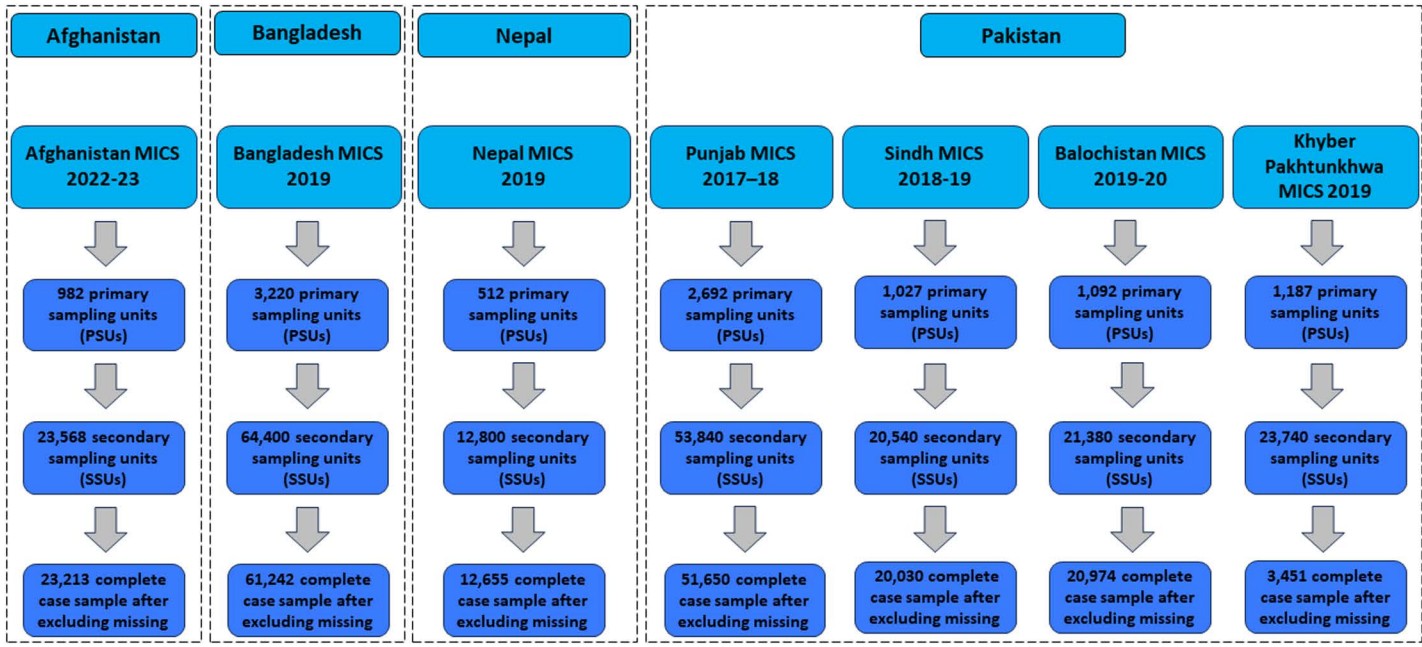

**Fig 1. Datasets from each of the countries used in this study.**

approval from the respective technical committee. We merged the four household-level provincial datasets of Pakistan to create the national-level dataset. After excluding missing observations, the complete case sample sizes for each of the four countries were as follows: Afghanistan (23,213), Bangladesh (61,242), Nepal (12,655) and Pakistan (96,105).

## Outcome variables

The study utilized the WASH variable as outcome, which assessed the overall status of household WASH facilities. This variable encompasses the combined status of drinking water, sanitation, and hygiene facilities based on the presence or absence of improved components across these three domains. Based on the Joint Monitoring Programme (JMP) 2017 's guide-lines, countries with or without improved WASH facilities are determined [25]. Drinking water facilities were categorized into improved sources, such as piped water into homes, yards, plots, neighbors, public taps, standpipes, protected spring, protected well, tube-well/borehole, rainwater harvesting, a cart with a tiny tank, water stand, sachet water, and bottled water and unimproved sources, including surface water, unprotected springs, unprotected wells, and other unprotected sources. Similarly, sanitation facilities were divided into improved types, such as piped sewer system, pit latrine, flush or pour flush to septic tank, or unknown loca-tions and unimproved types, including hanging toilets and latrines, pit latrines without slabs or open pits, open defecation (in a field, jungle, or other open spaces), open drains, and other forms of unimproved sanitation. Hygiene facilities were categorized based on the availability of a hand washing place, water, and soap, with improved facilities requiring all three compo-nents to be present. The WASH variable was categorized as improved when all three WASH domain were improved otherwise categorized as unimproved [6]. We also referred household having WASH facilities or coverage only when it was improved.

## Covariates

The covariates encompass a range of demographic, socioeconomic, and geographic variables. The demographic section includes age of the household head (young adult (aged below 35), middle-aged adult (aged 35 to 54), and older-aged adult (aged above 54)), sex of the household head (female or male), religion (Muslim, Hindu, and others), and ethnicity (Bengali or others) [26,27]. Due to the unavailability of data, we considered the ethnicity variable only for Ban-gladesh and the religion of household head variable for Bangladesh and Nepal's analyses. The socioeconomic section includes economic indicators such as education of the household head (pre-primary or no education, primary, secondary, higher), family size (small (1-5), medium (6-10), and large (10+)), access to mass media (with or without access), and wealth index quan-tiles (poor, middle, rich) [28]. If any of the three responses (radio, television, internet accessibil-ity) is yes, mass media accessibility is considered accessible otherwise inaccessible. Finally, the geographic section encompasses the place of residence (urban or rural) and the region variable represents the highest administrative division within each country: regions in Afghanistan (North, North East, East, South East, Central, Central Highland, West, and South) [29], divi-sions in Bangladesh (Mymensingh, Sylhet, Khulna, Dhaka, Chattogram, Rajshahi, Barishal, and Rangpur), provinces in Nepal (Sudurpashchim, Koshi or Province 1, Karnali, Lumbini, Mad-hesh or Province 2, Bagmati, and Gandaki), and provinces in Pakistan (Balochistan, Khyber Pakhtunkhwa, Punjab, and Sindh). The variables we analyzed in our study are shown in Fig 2.

## Statistical analyses

The statistical analyses of this study included several key steps. Initially to get the summary of the data we conducted the univariate and bivariate analyses. Then geographic mapping was

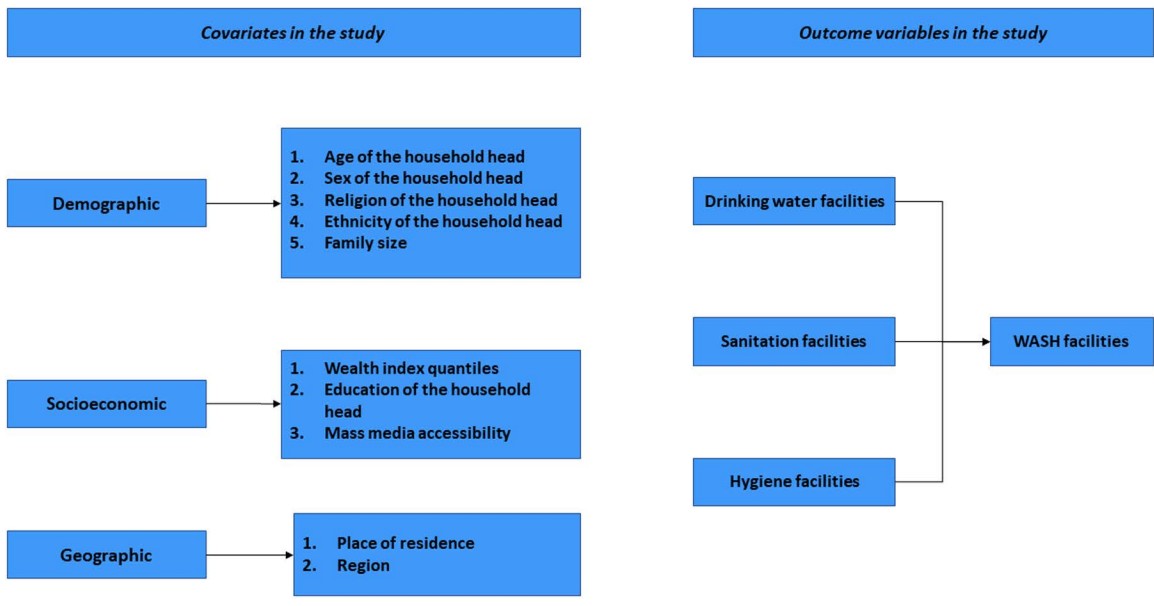

**Fig 2. Outcome variables and covariates of this study.**

used to illustrate variations in the WASH coverage across different regions in each country. Further, we used binary logistic regression to estimate the adjusted effects of the covariates. An adjusted odds ratio (aOR) plot was then used for offering crucial insights through a visual representation of the adjusted odds ratio of significant variables influencing the WASH facilities. Next the GVIF analysis was conducted to check the multicollinearity issues in the logistic regression models. Furthermore, Receiver Operating Characteristic (ROC) curve was used for the assessment of predictive performance of the models. This comprehensive methodology enabled a detailed investigation of the factors shaping WASH conditions in four diverse South Asian countries, identifying geographical, socioeconomic, and demographic differences. For data processing and analysis, we used the statistical software R (version 4.3.1) by R Core Team (2023) [30]. The analyses utilized several R packages, including tidyverse by Wickham *et al.* (2019) [31] for data manipulation and visualization, survey by Lumley (2024) [32] for handling complex survey data, and sf by Pebesma & Bivand (2023) [33] for spatial data processing. Additionally, pROC by Robin *et al.* (2011) [34] was used for ROC curve analysis, ggplot2 by Wickham *et al.* (2016) [35] for advanced graphical representation, and gridExtra by Auguie *et al.* (2017) [36] for arranging multiple plots in a grid format. Besides, we utilized the administrative boundary shape file of South Asian countries from an openly licensed database called geoBoundaries Global Administrative Database [37].

## Ethical considerations

The survey protocol for the datasets used in this study received ethical approval from the technical committees of the governments of Afghanistan, Bangladesh, Nepal, and Pakistan, overseen by their respective national statistical bureaus. In Afghanistan, the survey protocol was approved by the National Statistics and Information Authority (NSIA) technical committee in July 2022. In Bangladesh, the technical committee led by the Bangladesh Bureau of Statistics (BBS) approved the protocol. For Nepal, the Central Bureau of Statistics (CBS) approved the protocol as per the Statistical Act (1958) in September 2018. In Pakistan, the survey protocols

for Punjab, Sindh, Balochistan, and Khyber Pakhtunkhwa were approved by the Bureau of Statistics Institutional Review Board (IRB). In Punjab and Sindh, the protocol was approved in October 2017 [38]. Participants were informed that their involvement was voluntary and responses would remain confidential and anonymous. Besides they had the right to skip any questions, and could stop the interview at any time. As the data used in this study were sourced from secondary datasets, no further ethical approval was necessary.

## Results

### Country-wise coverage of WASH facilities

The outcome variable WASH facilities encompass the combined status of drinking water, sanitation, and hygiene facilities based on the presence or absence of improved components across these three domains. Table 1 shows the prevalence of each component across the four countries.

Out of the four countries we studied, Bangladesh had the highest coverage of improved drinking water facilities at the household level (98.56%). The next three countries were Nepal (96.95%), Pakistan (94.47%), and Afghanistan had the lowest percentage of all four (73.70%). Next, Nepal had the largest household level coverage of improved sanitation facilities (94.06%). Bangladesh came next with 84.60%, Pakistan stands next with 73.16%, and Afghanistan had the lowest percentage of all four with 65.57%. In terms of improved hygiene facilities, Nepal had the highest coverage rate of 80.28%. Pakistan had 75.28% after that. Among the four, Afghanistan had the lowest percentage at 55.15%, followed by Bangladesh at 56.28%.

Fig 3 illustrates the distribution of country-wise household level coverage of WASH facilities. Out of the four countries we studied, Nepal had the highest household level coverage of WASH facilities, at 75.33%. After that, Pakistan has 59.47%, and Bangladesh had 50.28%. Finally, with only 33.54% WASH facilities, Afghanistan had the lowest percentage of the four.

### Region-wise coverage of WASH facilities for four countries

The mapping (Fig 4) illustrates the region-wise household-level coverage of WASH facilities in four South Asian countries. This map demonstrates that there exist significant geographical differences in the coverage of WASH facilities among the four nations under study: Bangladesh, Pakistan, Nepal, and Afghanistan. Nepal's Koshi, Bagmati, Gandaki, and Lumbini regions did better, maintaining above 80% household-level coverage of WASH facilities. The Punjab region of Pakistan and the Madesh region of Nepal have also had better results, maintaining coverage levels of almost 66-69%. Next, with nearly or above 56-63% WASH facilities, the Rangpur, Khulna, and Rajshahi regions of Bangladesh and the Sindh Province of Pakistan stand. Then, with coverage of nearly 50-54%, the Karnali and Sudurpashchim provinces of Nepal, the Central region of Afghanistan, and the Dhaka region of Bangladesh stand. Ranging from 40-49% in the Mymensingh, Chattogram, and Sylhet regions of Bangladesh, the South-East region of Afghanistan, and Pakistan's Balochistan region had comparatively lower coverage. Then, with coverage of nearly 30-39%, the Khyber Pakhtunkhwa Province of Pakistan

Table 1. Country-wise household level coverage of WASH components.

| Country (Year) | Drinking water facilities | Sanitation facilities | Hygiene facilities | WASH facilities |
|---|---|---|---|---|
| Afghanistan (2022) | 73.70% | 65.57% | 55.15% | 33.54% |
| Bangladesh (2019) | 98.56% | 84.60% | 56.28% | 50.28% |
| Nepal (2019) | 96.95% | 94.06% | 80.28% | 75.33% |
| Pakistan (2017-19) | 94.47% | 73.16% | 75.28% | 59.47% |

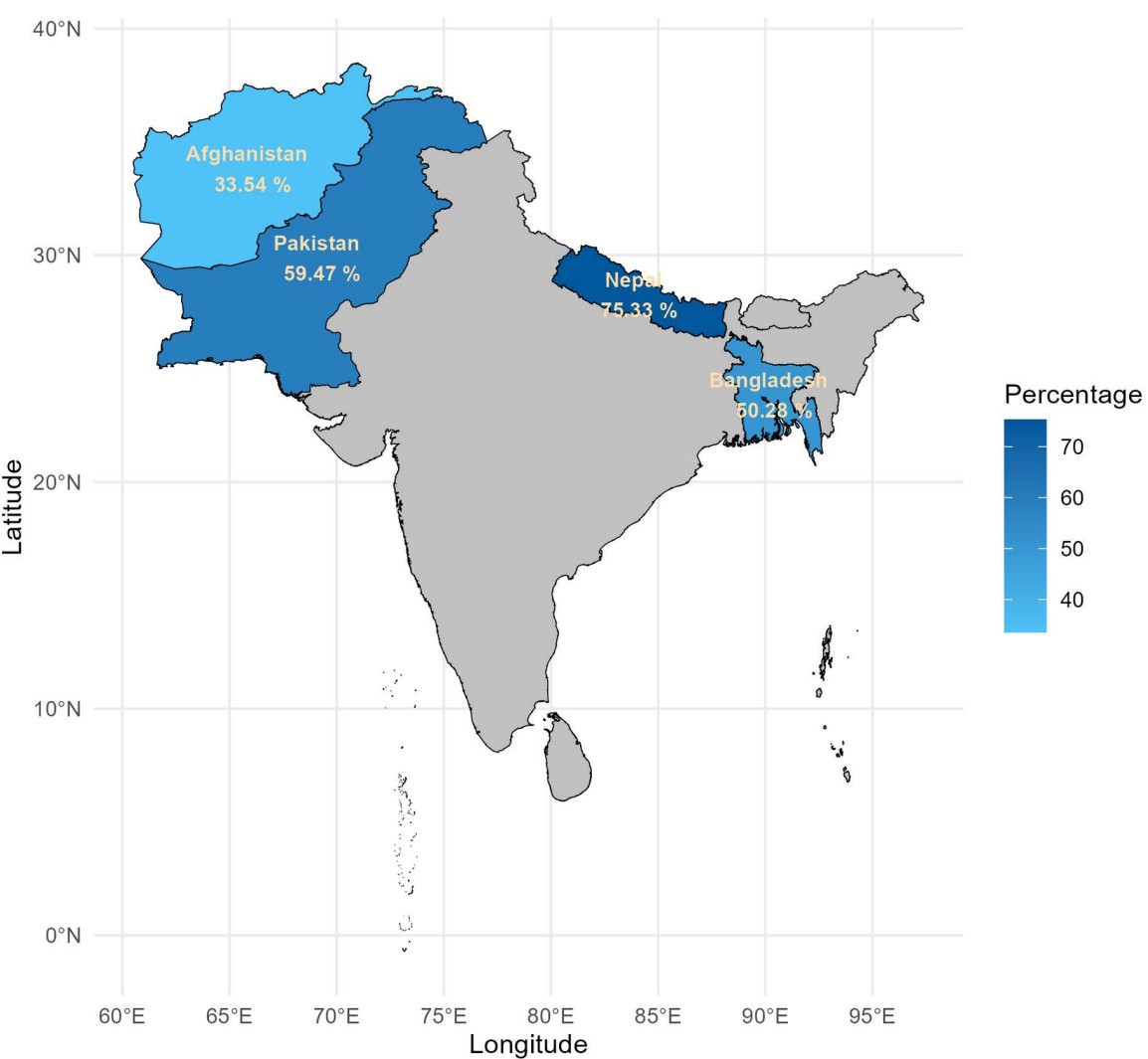

**Fig 3. Country-wise household level coverage of WASH facilities.**

and the North and West regions of Afghanistan performed badly. Afghanistan's North East, East, Central Highland, and South regions and the Barisal region of Bangladesh ranked the worst, with coverage below 30%. Therefore, Nepal did quite well in terms of household-level WASH facility coverage among our four study nations, followed by Pakistan, while Afghanistan and Bangladesh did poorly.

## Distribution of WASH facilities by demographic, socio-economic, and geographic factors

The percentage distribution of households with WASH facilities in four South Asian countries is illustrated in Table 2. Urban households across four countries showed a higher percentage of WASH facilities compared to rural one. Also, when the level of economic status rises, the percentage of households with WASH facilities rises across all countries. Afghanistan and Bangladesh had higher (3%) coverage rates for WASH facilities among households headed by males. However, households headed by females in Pakistan, and Nepal have a 1-7% higher

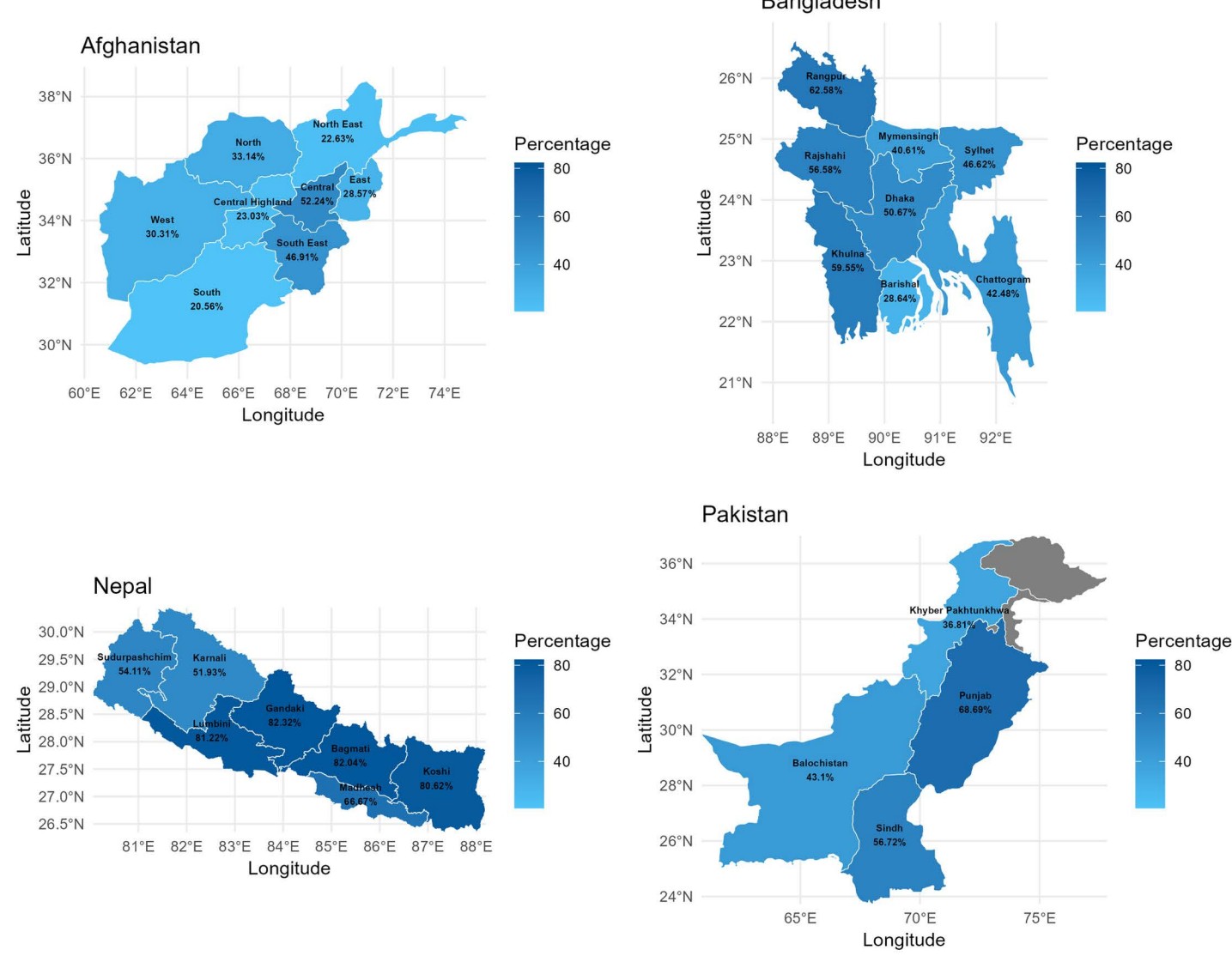

**Fig 4. Region-wise household level coverage of WASH facilities.**

coverage rate of WASH facilities. The sex of household heads was significantly associated with WASH facilities at a 5% level of significance for Pakistan and Bangladesh, but for Nepal and Afghanistan, it was not significantly associated. For the middle-aged adult household head, the coverage of WASH facilities was higher in Nepal and Bangladesh, but in Pakistan and Afghanistan, for the older-aged adult household head, the coverage of WASH facilities was higher than the other age group. As education levels increased, each country in the study saw a rise in access to WASH facilities. Muslims in Bangladesh had the best access to WASH facilities, followed by Hindus and other religions. But in Nepal, the Hindu community had the best access to WASH facilities. Compared to other ethnic communities, the Bengali community in Bangladesh had a 21% higher household-level coverage of WASH. In Nepal and Bangladesh, the coverage of WASH was high for large family sizes, whereas in Pakistan and Afghanistan the opposite was true. Households with media access had the highest coverage of WASH facilities across all of

**Table 2. The frequency distribution of WASH facilities by demographic, socio-economic, and geographic factors.**

| Variables | Household having Improved WASH Facilities | | | |
|---|---|---|---|---|
| | Afghanistan (2022) | Bangladesh (2019) | Nepal (2019) | Pakistan (2017-19) |
| | N (%) | N (%) | N (%) | N (%) |
| Place of residence | *** | *** | *** | *** |
| Urban | 3808 (60.02) | 8494 (62.62) | 6841 (79.64) | 47655 (79.73) |
| Rural | 3977 (23.58) | 22299 (46.77) | 2691 (66.20) | 17132 (47.15) |
| Economic status | *** | *** | *** | *** |
| Poor | 1008 (10.34) | 7584 (29.89) | 2807 (56.34) | 12353 (30.70) |
| Middle | 1303 (29.28) | 5983 (50.30) | 1896 (79.35) | 12362 (67.88) |
| Rich | 5473 (60.74) | 17225 (71.85) | 4830 (91.42) | 32434 (86.13) |
| Sex of household head | | *** | | *** |
| Male | 7410 (33.72) | 27067 (50.63) | 6845 (75.15) | 51396 (58.89) |
| Female | 377 (30.39) | 3726 (47.88) | 2688 (75.78) | 5755 (65.17) |
| Age of household head | | *** | *** | *** |
| Young adult | 2176 (32.00) | 5779 (44.40) | 2080 (72.25) | 9462 (52.08) |
| Middle-aged adult | 3798 (34.14) | 15858 (52.14) | 4743 (76.83) | 52458 (59.97) |
| Older-aged adult | 1811 (34.24) | 9158 (51.42) | 27.09 (75.22) | 16227 (63.69) |
| Education of household head | *** | *** | *** | *** |
| None or Pre-primary | 3637 (25.01) | 8135 (37.91) | 2875 (63.68) | 20884 (45.62) |
| Primary | 1135 (39.25) | 7315 (44.10) | 3252 (76.89) | 8134 (61.06) |
| Secondary | 1952 (48.75) | 9231 (58.95) | 2640 (85.72) | 17985 (73.45) |
| Higher | 1059 (59.98) | 6114 (81.12) | 765 (92.19) | 10148 (81.06) |
| Religion of household head | | *** | | |
| Muslim | – | 27962 (50.60) | 241 (67.35) | – |
| Hindu | – | 2557 (48.82) | 8122 (75.65) | – |
| Others | – | 274 (36.83) | 1170 (74.97) | – |
| Ethnicity | | *** | | |
| Bengali | – | 30584 (50.53) | – | – |
| Others | – | 212 (29.71) | – | – |
| Family size | * | *** | * | *** |
| Small | 1987 (36.04) | 24851 (50.05) | 7488 (75.91) | 22306 (62.62) |
| Medium | 4255 (33.11) | 5631 (50.64) | 1921 (72.84) | 28840 (57.81) |
| Large | 1543 (31.83) | 312 (66.24) | 125 (81.09) | 6002 (56.64) |
| Mass media accessibility | *** | *** | *** | *** |
| With access | 5974 (45.75) | 23042 (60.74) | 7866 (82.87) | 47013 (71.46) |
| Without access | 1810 (17.83) | 7754 (33.27) | 1667 (52.69) | 10138 (33.44) |
| Region | *** | *** | *** | *** |
| *Afghanistan-* | | | | |
| North | 1164 (33.14) | – | – | – |
| North East | 741 (22.63) | – | – | – |
| East | 651 (28.57) | – | – | – |
| South East | 820 (46.91) | – | – | – |
| Central | 2413 (52.24) | – | – | – |
| Central Highland | 203 (23.03) | – | – | – |
| West | 1165 (30.31) | – | – | – |
| South | 628 (20.56) | – | – | – |

*(Continued)*

**Table 2.** (Continued)

| Variables | Household having Improved WASH Facilities | | | |
|---|---|---|---|---|
| | Afghanistan (2022) | Bangladesh (2019) | Nepal (2019) | Pakistan (2017-19) |
| | N (%) | N (%) | N (%) | N (%) |
| *Bangladesh-* | | | | |
| Sylhet | – | 1709 (46.42) | – | – |
| Rangpur | – | 4837 (62.58) | – | – |
| Rajshahi | – | 4948 (56.58) | – | – |
| Mymenshing | – | 1852 (40.61) | – | – |
| Khulna | – | 4341 (59.55) | – | – |
| Dhaka | – | 7860 (50.67) | – | – |
| Chattogram | – | 4561 (42.48) | – | – |
| Barishal | – | 999 (28.64) | – | – |
| *Nepal-* | | | | |
| Sudurpashchim | – | – | 548 (54.11) | – |
| Karnali | – | – | 348 (51.93) | – |
| Lumbini | – | – | 1731 (81.22) | – |
| Bagmati | – | – | 2708 (82.04) | – |
| Koshi | – | – | 1793 (80.62) | – |
| Gandaki | – | – | 1023 (82.32) | – |
| Madhesh | – | – | 1382 (66.67) | – |
| *Pakistan-* | | | | |
| Balochistan | – | – | – | 9040 (43.10) |
| Khyber Paktunkhwa | – | – | – | 1270 (36.81) |
| Punjab | – | – | – | 35478 (68.69) |
| Sindh | – | – | – | 11361 (56.72) |
| Overall | 23213 (33.54) | 61242 (50.28) | 12655 (75.33) | 96105 (59.47) |

Pearson's Chi-square test of association significance: *** p value < 0.001, ** p value < 0.01, * p value < 0.05.

our study countries. In Bangladesh, Rangpur had the highest household coverage of WASH facilities (62.58%), while Barishal had the lowest coverage (28.64%). In Afghanistan, the South, North East and Central Highland region had the lowest (20-23.03%) household-level coverage of WASH, while the South East and Central region had the highest coverage (46.91-52.24%). In Nepal, Gandaki had the highest household coverage of WASH facilities (82.32%), while Karnali had the lowest coverage (51.93%). Punjab had the highest household coverage of WASH facilities in Pakistan (68.69%), while Khyber Pakhtunkhwa had the lowest coverage (36.81%).

## Binary logistic regression model

Table 3 displays the findings of a binary logistic regression model using WASH facilities as the response variable for each of the four study countries. Rural households were associated with significantly lower odds of having WASH facilities, with 46% lower odds in Afghanistan (aOR = 0.54; 95% CI: 0.44–0.65; $p < 0.001$) and 30% lower odds in Pakistan (aOR: 0.70; 95% CI: 0.64–0.77; $p < 0.001$). In all countries, wealthier households had significantly higher odds of having better access to WASH facilities. In Afghanistan, the wealthiest household were associated with 7.83 times (aOR:7.83; 95% CI: 6.58–9.32; $p < 0.001$) odds of having better access to WASH facilities than the poor household. In Bangladesh, Nepal, and Pakistan, the odds were 5.75 times (aOR: 5.75; 95% CI: 5.34–6.20; $p < 0.001$), 5.80 times (aOR: 5.80;

**Table 3. Binary logistic regression model adjusted for demographic, socio-economic, and geographic factors with WASH facilities as outcome.**

| Variables | Household having WASH facilities | | | |
| --- | --- | --- | --- | --- |
| | Afghanistan (2022) | Bangladesh (2019) | Nepal (2019) | Pakistan (2017-19) |
| | aOR (95% CI) | aOR (95% CI) | aOR (95% CI) | aOR (95% CI) |
| Place of residence | | | | |
| Urban (ref.) | 1.00 | 1.00 | 1.00 | 1.00 |
| Rural | 0.54 (0.44, 0.65) *** | 1.06 (0.97, 1.15) | 0.93 (0.77, 1.13) | 0.70 (0.64, 0.77) *** |
| Economic status | | | | |
| Poor (ref.) | 1.00 | 1.00 | 1.00 | 1.00 |
| Middle | 3.50 (2.60, 3.58) *** | 2.21 (2.08, 2.32) *** | 2.77 (2.33, 3.30) *** | 4.09 (3.85, 4.34) *** |
| Rich | 7.83 (6.58, 9.32) *** | 5.75 (5.34, 6.20) *** | 5.80 (4.52, 7.44) *** | 9.64 (8.79, 10.58) *** |
| Sex of household head | | | | |
| Male (ref.) | 1.00 | 1.00 | 1.00 | 1.00 |
| Female | 1.13 (0.92, 1.38) | 0.96 (0.90, 1.02) | 1.09 (0.94, 1.26) | 1.16 (1.09, 1.24) *** |
| Age of household head | | | | |
| Young adult (ref.) | 1.00 | 1.00 | 1.00 | 1.00 |
| Middle-aged adult | 1.10 (0.98, 1.24) | 1.50 (1.42, 1.59) *** | 1.78 (1.54, 2.05) *** | 1.17 (1.11, 1.22) *** |
| Older-aged adult | 1.12 (0.98, 1.28) | 1.71 (1.60, 1.83) *** | 2.17 (1.84, 2.57) *** | 1.27 (1.20, 1.35) *** |
| Education of household head | | | | |
| None or Pre-primary (ref.) | 1.00 | 1.00 | 1.00 | 1.00 |
| Primary | 1.19 (1.03, 1.37) * | 1.17 (1.11, 1.24) *** | 1.57 (1.37, 1.79) *** | 1.24 (1.17, 1.31) *** |
| Secondary | 1.55 (1.36, 1.78) *** | 1.69 (1.59, 1.80) *** | 2.03 (1.71, 2.41) *** | 1.51 (1.43, 1.59) *** |
| Higher | 1.88 (1.55, 2.87) *** | 3.66 (3.33, 4.01) *** | 2.96 (1.83, 4.80) *** | 1.67 (1.55, 1.81) *** |
| Religion of household head | | | | |
| Muslim (ref.) | – | 1.00 | 1.00 | – |
| Hindu | – | 0.81 (0.72, 0.91) ** | 1.50 (0.91, 2.47) | – |
| Others | – | 1.07 (0.66, 1.72) ** | 1.37 (0.80, 2.34) | – |
| Ethnicity | | | | |
| Bengali (ref.) | – | 1.00 | – | – |
| Others | – | 0.92 (0.56, 1.52) | – | – |
| Family size | | | | |
| Small (ref.) | 1.00 | 1.00 | 1.00 | 1.00 |
| Medium | 0.86 (0.75, 0.97) * | 1.06 (1.01, 1.13) * | 1.08 (0.96, 1.22) | 1.00 (0.96, 1.04) |
| Large | 0.72 (0.62, 0.84) *** | 1.61 (1.27, 2.05) *** | 1.32 (0.83, 2.12) | 1.03 (0.95, 1.10) |
| Mass media accessibility | | | | |
| Without access (ref.) | 1.00 | 1.00 | 1.00 | 1.00 |
| With access | 1.09 (0.97, 1.23) | 1.35 (1.28, 1.42) *** | 1.79 (1.58, 2.03) *** | 1.51 (1.44, 1.59) *** |
| Region | | | | |
| *Afghanistan-* | | | | |
| North (ref.) | 1.00 | – | – | – |
| North East | 0.67 (0.50, 0.90) ** | – | – | – |
| East | 1.09 (0.80, 1.49) | – | – | – |
| South East | 2.36 (1.80, 3.09) *** | – | – | – |
| Central | 1.02 (0.79, 1.30) | – | – | – |
| Central Highland | 0.75 (0.56, 1.01) | – | – | – |
| West | 0.96 (0.69, 1.34) | – | – | – |
| South | 0.63 (0.47, 0.85) ** | – | – | – |

*(Continued)*

**Table 3.** (Continued)

| Variables | Household having WASH facilities | | | |
| --- | --- | --- | --- | --- |
| | Afghanistan (2022) | Bangladesh (2019) | Nepal (2019) | Pakistan (2017-19) |
| | aOR (95% CI) | aOR (95% CI) | aOR (95% CI) | aOR (95% CI) |
| Bangladesh- | | | | |
| Barishal (ref.) | – | 1.00 | – | – |
| Chattogram | – | 1.09 (0.99, 1.21) | – | – |
| Dhaka | – | 1.30 (1.16, 1.46) *** | – | – |
| Khulna | – | 3.14 (2.82, 3.49) *** | – | – |
| Mymenshing | – | 1.96 (1.73, 2.22) *** | – | – |
| Rajshahi | – | 3.42 (3.08, 3.81) *** | – | – |
| Rangpur | – | 6.12 (5.51, 6.80) *** | – | – |
| Sylhet | – | 1.77 (1.53, 2.05) *** | – | – |
| *Nepal-* | | | | |
| Koshi (ref.) | – | – | 1.00 | |
| Madhesh | – | – | 0.34 (0.24, 0.49) *** | – |
| Bagmati | – | – | 0.49 (0.36, 0.67) *** | – |
| Gandaki | – | – | 0.81 (0.58, 1.12) | – |
| Lumbini | – | – | 0.99 (0.71, 1.39) | – |
| Karnali | – | – | 0.43 (0.32, 0.59) *** | – |
| Sudurpashchim | – | – | 0.30 (0.21, 0.43) *** | – |
| *Pakistan-* | | | | |
| Balochistan (ref.) | – | – | – | 1.00 |
| Khyber Paktunkhwa | – | – | – | 2.39 (2.02, 2.82) *** |
| Punjab | – | – | – | 3.48 (3.13, 3.86) *** |
| Sindh | – | – | – | 1.54 (1.37, 1.72) *** |

Wald test significance:

***$p$ value < 0.001, **$p$ value < 0.01, *$p$ value < 0.05.

95% CI: 4.52–7.44; $p < 0.001$), and 9.64 times (aOR: 9.64; 95% CI: 8.79–10.58; $p < 0.001$), respectively. Households with older-aged adult were associated of having WASH facilities compared to the young adult household head, with respectively 71% higher odds in Bangladesh (aOR: 1.71; 95% CI: 1.60–1.83; $p < 0.001$), 2.17 times odds in Nepal (aOR: 2.17; 95% CI: 1.84–2.57; $p < 0.001$), and 27% higher odds in Pakistan (aOR: 1.27; 95% CI: 1.20–1.35; $p < 0.001$). The odds of having better WASH facilities were higher in all countries for those had higher education levels. In Afghanistan, the household head with higher education were associated with 88% higher (aOR: 1.88; 95% CI: 1.55–2.87; $p < 0.001$) odds of having better access to WASH than the household head with no education. In Bangladesh, Nepal, and Pakistan, the odds were 3.66 times (aOR: 3.66; 95% CI: 3.33–4.01; $p < 0.001$), 2.96 times (aOR: 2.96; 95% CI: 1.83–4.80; $p < 0.001$), and 67% higher (aOR: 1.67; 95% CI: 1.55–1.81; $p < 0.001$), respectively. Household with access to mass media were associated with higher odds of having WASH facilities except Afghanistan. Across all of the study countries, region was significantly associated with WASH facilities. In Bangladesh, households in the Rangpur region were associated with 6.12 times (aOR: 6.12; 95% CI: 5.51–6.80; $p < 0.001$) odds of having WASH facilities than those in the Barishal region, whereas households in the Dhaka region were associated with 30% higher odds (aOR: 1.30; 95% CI: 1.16–1.46; $p < 0.001$). In Afghanistan, households in the South East were associated with 2.36 times (aOR: 2.36; 95%

CI: 1.80–3.09; $p < 0.001$) odds of having improved WASH facilities than those in the North, while households in the South region were associated with 37% lower (aOR: 0.63; 95% CI: 0.47–0.85; $p < 0.01$) odds compared to the North region. In Pakistan, households in the Punjab area were associated with 3.48 times (aOR: 3.48; 95% CI: 3.13–3.86; $p < 0.001$) the odds of having better access to WASH than households in the Balochistan region, whereas households in the Sindh region were associated with 54% higher (aOR: 1.54; 95% CI: 1.37–1.72; $p < 0.001$) odds compared to the Balochistan region. In Nepal, households in the Madhesh were associated with 66% lower (aOR: 0.34; 95% CI: 0.24–0.49; $p < 0.001$) odds of having better access to WASH than households in the Koshi region, whereas households in the Sudurpashchim region were associated with 70% lower (aOR: 0.30; 95% CI: 0.21–0.43; $p < 0.001$) odds compared to the Koshi region.

The adjusted odds ratio plot [39] of household-level coverage of WASH facilities in four South Asian countries is illustrated in Fig 5, which provides a visual representation of significant variable levels and their odds of having WASH facilities compared to the baseline level for each country.

Table 4 presents the generalized variance inflation factors (GVIFs) analysis for the binary logistic regression models to assess multicollinearity. It was observed that the regression models did not have multicollinearity issues, as all squared adjusted GVIF values were below five [40,41]. Further detail was provided in the supplementary section (S1 Table and S2 Table).

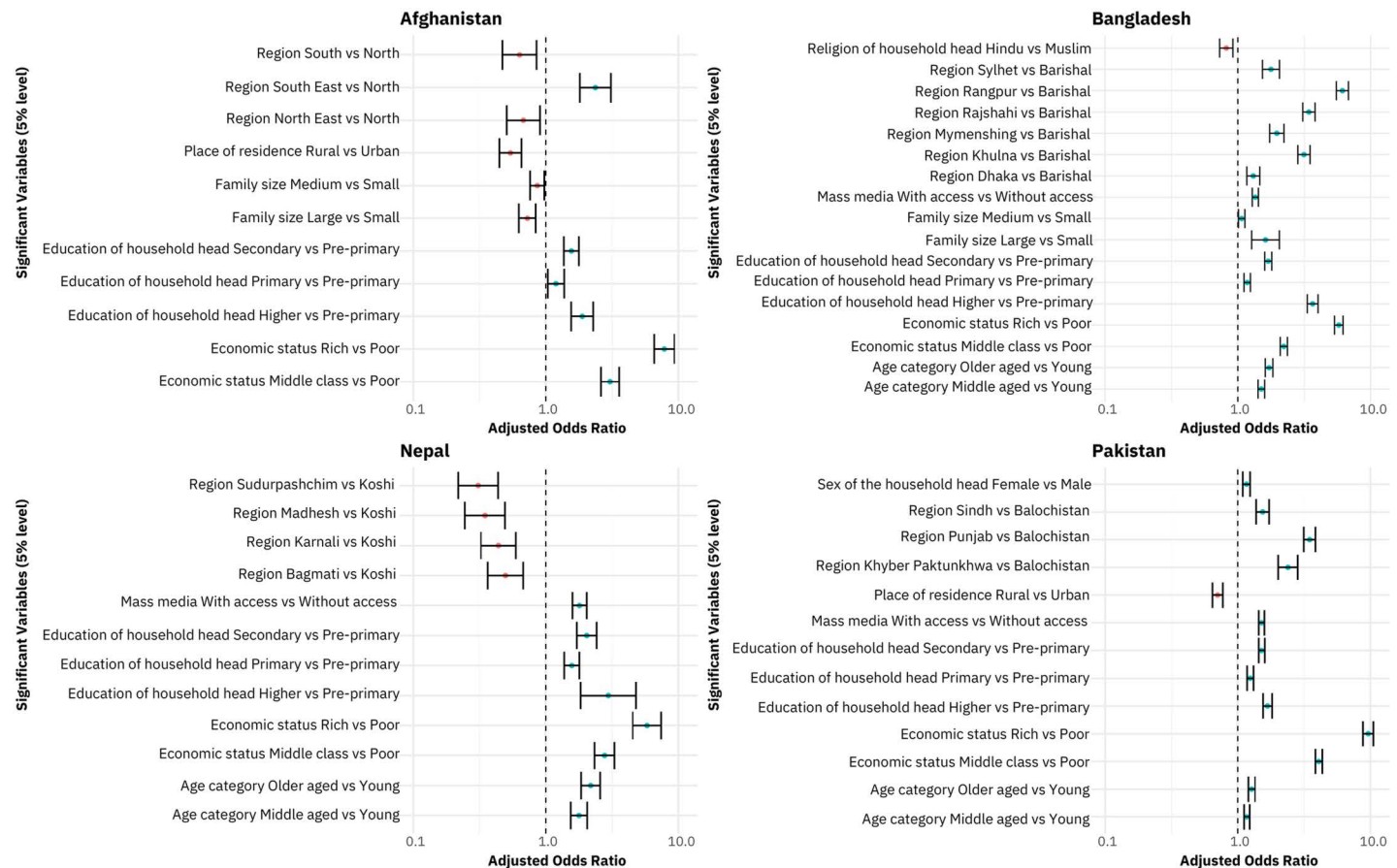

**Fig 5. Adjusted Odds Ratio plot of household level coverage of WASH facilities in four South Asian countries.**

## Sensitivity analysis

Fig 6 illustrates the Receiver Operating Characteristic (ROC) curve, presenting the classification performance of the logistic regression model for WASH facilities across four South Asian countries. An AUC of 1 indicates perfect classification of samples, while an AUC of 0.5 suggests the model performs no better than random chance. Scores above 0.7 are regarded as good model performance [42]. The Area Under the ROC Curve (AUC) values are 0.79 for Afghanistan, 0.78 for Bangladesh, 0.78 for Nepal, and 0.83 for Pakistan. Therefore, these AUC values suggest that the models have good discriminating power in classifying WASH facilities outcomes [41].

## Discussion

Our study investigated the significant demographic, socioeconomic and regional factors that influence the coverage of WASH facilities across four South Asian countries, revealing substantial disparities. The findings are consistent with patterns observed globally, as highlighted by Rahut *et al.* (2022) [39], which examined WASH coverage across 42 developing countries. Research showed that access to clean water, sanitation, and hygiene is still a major challenge in LMICs, especially in regions faces economic and geographic challenges. For example, studies from sub-Saharan Africa revealed that although water access has improved, sanitation still lags, especially in areas struggling with poverty [26].

One of the major findings of this study was the existing regional disparity in WASH coverage across the four study countries. Nepal showed an overall better WASH coverage (75.33%), while Afghanistan showed much lower rate (33.54%). This disparity was largely attributed to Afghanistan's ongoing conflict and severe environmental challenges [43]. Additionally, disparities exist within countries, with some regions performing significantly better than others. Geographic factors such as mountainous terrain, remote and arid regions responsible uneven distribution of WASH coverage [44,45]. The difficulties in building and maintaining WASH facilities in these challenging areas result in lower coverage compared to more accessible regions. For example, all regions of Afghanistan, except Central and South East, Barisal and Mymensingh in Bangladesh, Sudurpashchim and Karnali in Nepal, and Khyber-Pakhtunkhwa and Balochistan in Pakistan lag behind compared to other areas of respective countries. This is consistent with findings from Southern Africa, where environmental conditions significantly impact WASH improvements [46]. Further, our study found that urban households generally

**Table 4. GVIF for binary logistic regression model adjusted for demographic, socio-economic, and geographic factors with WASH facilities as outcome.**

| Variables | Squared Adjusted GVIF values | | | |
|---|---|---|---|---|
| | Afghanistan (2022) | Bangladesh (2019) | Nepal (2019) | Pakistan (2017-19) |
| Place of residence | 1.58 | 1.08 | 1.2 | 1.59 |
| Economic status | 1.35 | 1.31 | 1.3 | 1.31 |
| Sex of household head | 1.1 | 1.05 | 1.38 | 1.13 |
| Age of household head | 1.19 | 1.09 | 1.23 | 1.07 |
| Education of household head | 1.11 | 1.11 | 1.18 | 1.09 |
| Religion of household head | – | 1.67 | 1.12 | – |
| Ethnicity | – | 2.59 | – | – |
| Family size | 1.19 | 1.07 | 1.14 | 1.07 |
| Mass media accessibility | 1.27 | 1.35 | 1.2 | 1.09 |
| Region | 1.12 | 1.06 | 1.11 | 1.11 |

possess better WASH coverage compared to rural ones. This urban-rural division highlights the advantages of infrastructure and service availability in urban areas, and the challenges faced by rural areas. Our regional and urban-rural inequalities finding got additional validation from the previous studies conducted in Nepal [26,44], Bangladesh [6], and studies in global level [39,47,48].

Our study highlighted significant socioeconomic disparities in WASH coverage, with wealthier households securing better access compared to poorer ones. This finding was consistent with previous research, such as Armah *et al.* (2018) [26] found that in sub-Saharan Africa higher-income households had better access to sanitation and water services. Similarly, in Bangladesh, wealthier families were more likely to have safe drinking water and adequate sanitation [6]. Further, our study found that WASH coverage improved with higher levels of education. In Izmir, Turkey, study showed individuals with at least eight years of education had better WASH access compared to those with less education [17]. Global level studies also support these findings, with Rahut *et al.* (2022) [39] noting that socioeconomic factors play role in WASH disparities. Research in Pakistan also highlighted that household wealth status and education of household head significantly impacts WASH access [24]. Furthermore,

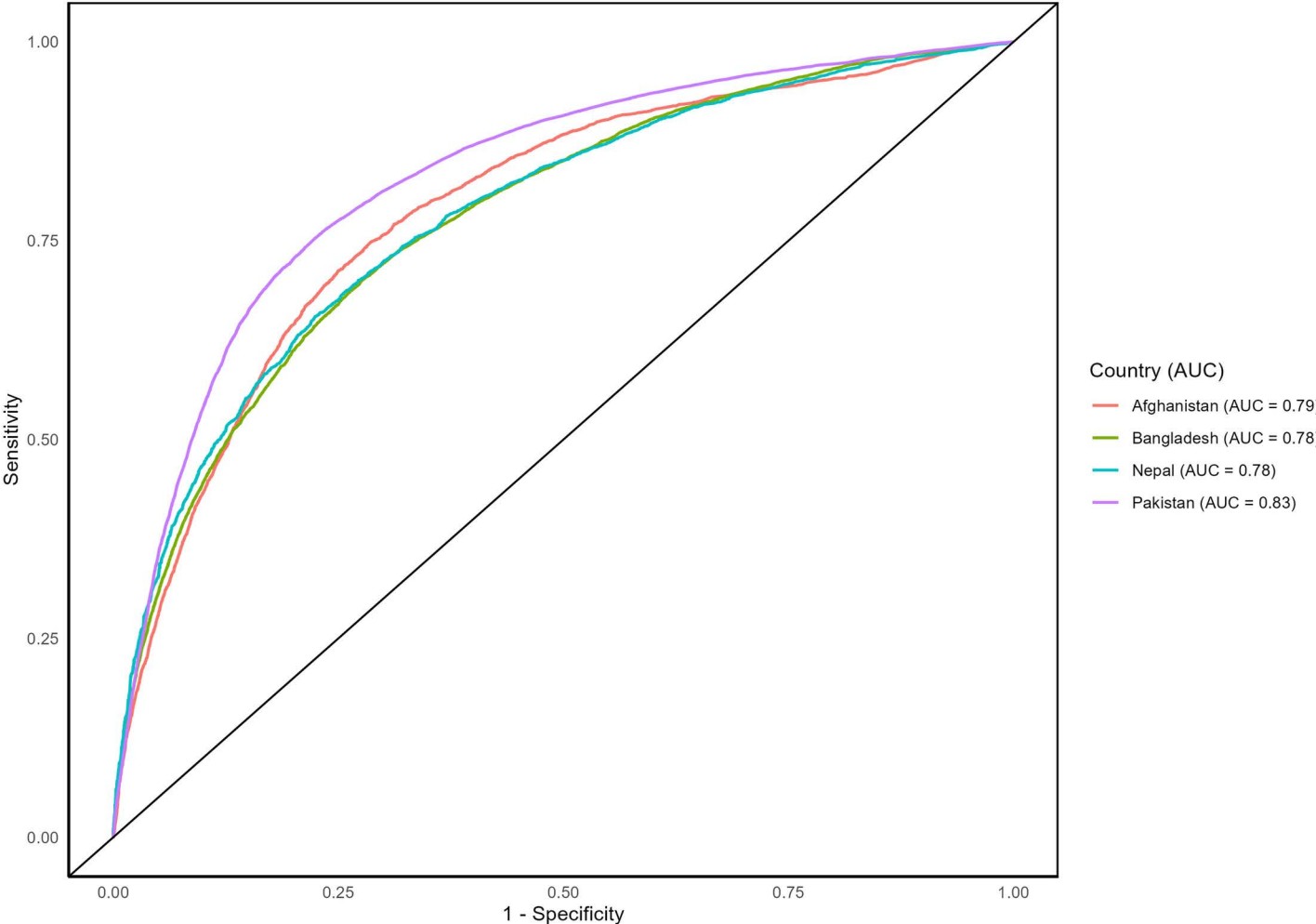

**Fig 6. ROC curve and AUC for logistic regression model of WASH facilities in four South Asian countries.**

access to mass media was associated with higher odds of having WASH facilities, consistent with findings from Tanzania that underscore the role of media accessibility in enhancing WASH coverage [49].

Demographic factors significantly influenced WASH coverage across the study countries. Households led by older-aged adults generally had better WASH access compared to those led by younger adults. This finding aligns with a study in Ethiopia, which also highlighted the significant association between the age of the household head and WASH access [2]. Family size had varied effects on WASH coverage: it was significantly associated with WASH access in Afghanistan and Bangladesh, but not in Pakistan and Nepal. This variation aligns with previous studies, which found significant associations between WASH and family size in Bangladesh [6] but non-significant in Ethiopia [2]. Additionally, households with a female head in Pakistan had higher odds of having WASH facilities, this was not observed in other study countries. Similar findings were reported in Ethiopia, where female-headed households also had better WASH access [2].

Ethnic disparities were also evident in Bangladesh, with the Bengali community showed notably higher household-level WASH coverage compared to other ethnic groups. A previous study also highlighted ethnic disparities in WASH access of Bangladesh [27].

Our study aims to contribute to the policy of the Human Right to Water and Sanitation (HRtWS), which ensures that everyone has the right to safe, clean water and sanitation [50]. To ensure this, it is essential to consider Sustainable WASH principles at national level. Which includes technical, institutional, environmental, financial, and social sustainability. Technical sustainability focuses on using appropriate and maintainable technologies, while institutional sustainability involves strengthening governance and policy frameworks. Environmental sustainability ensures that WASH practices are ecologically responsible, financial sustainability emphasizes local funding mechanisms, and social sustainability ensures inclusivity and equity in WASH services [51]. Integrating these principles can enhance the effectiveness of WASH interventions.

We analyzed nationally representative data for each of the study countries. This enhances the credibility of the findings and allows for meaningful comparisons between regions. However, our study had several limitations. Our analyses are based on secondary datasets, and there may be potential recall bias due to data collected from household member responses. Further, the ability to assess changes over time or causality was not possible due to the cross-sectional nature of data. Despite these limitations, our study provides the overview of the WASH conditions in our study countries, where such studies were limited.

## Conclusion

Our study offers a general overview of the WASH conditions and highlights significant disparities in WASH coverage based on demographic, socioeconomic and regional factors in our four study countries. Regional variations were evident from spatial mapping, with remote areas such as those with mountainous terrain, arid conditions, or poor communication experiencing the most challenges. Logistic regression analysis further revealed that factors such as place of residence, access to media, age of the household head, and family size significantly impact WASH access. Poorer and less educated households had the limited access, while urban, educated and wealthier households had better WASH facilities. To address these inequalities, policymakers should prioritize interventions in remote areas, implement targeted subsidies for WASH equipment, and conduct awareness campaigns. These strategies are essential for improving WASH coverage and advancing towards achieving SDG 6 of ensuring everyone's access to reliable water and sanitation services by 2030.

## Supporting information

**S1 Table. GVIF for binary logistic regression model adjusted for demographic, socio-economic, and geographic factors with WASH facilities as outcome in Afghanistan and Bangladesh.**
(DOCX)

**S2 Table. GVIF for binary logistic regression model adjusted for demographic, socio-economic, and geographic factors with WASH facilities as outcome in Nepal and Pakistan.**
(DOCX)

## Author contributions

**Conceptualization:** Md. Hasibul Islam Jitu, Mohammad Shahed Masud.

**Data curation:** Md. Hasibul Islam Jitu.

**Formal analysis:** Md. Hasibul Islam Jitu.

**Methodology:** Md. Hasibul Islam Jitu, Mohammad Shahed Masud.

**Supervision:** Mohammad Shahed Masud.

**Validation:** Md. Hasibul Islam Jitu, Mohammad Shahed Masud.

**Visualization:** Md. Hasibul Islam Jitu, Mohammad Shahed Masud.

**Writing – original draft:** Md. Hasibul Islam Jitu.

**Writing – review & editing:** Md. Hasibul Islam Jitu, Mohammad Shahed Masud.

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
