## [Decision Letter · Decision Letter 0]

23 Aug 2024

PONE-D-24-27103Demographic, Socioeconomic and Regional Disparities in the Coverage of Water, Sanitation and Hygiene (WASH) Facilities in Four South Asian CountriesPLOS ONE

Dear Dr. Jitu,

Thank you for submitting your manuscript to PLOS ONE. After careful consideration, we feel that it has merit but does not fully meet PLOS ONE’s publication criteria as it currently stands. Therefore, we invite you to submit a revised version of the manuscript that addresses the points raised during the review process.

We look forward to receiving your revised manuscript.

Kind regards,

Ramesh Adhikari

Academic Editor

PLOS ONE

Journal Requirements:

2. We note that Figures 3 and 4 in your submission contain [map/satellite] images which may be copyrighted. All PLOS content is published under the Creative Commons Attribution License (CC BY 4.0), which means that the manuscript, images, and Supporting Information files will be freely available online, and any third party is permitted to access, download, copy, distribute, and use these materials in any way, even commercially, with proper attribution. For these reasons, we cannot publish previously copyrighted maps or satellite images created using proprietary data, such as Google software (Google Maps, Street View, and Earth). For more information, see our copyright guidelines: http://journals.plos.org/plosone/s/licenses-and-copyright.

a. You may seek permission from the original copyright holder of Figures 3 and 4 to publish the content specifically under the CC BY 4.0 license.  

Reviewers' comments:

Reviewer's Responses to Questions

**Comments to the Author**

1. Is the manuscript technically sound, and do the data support the conclusions?

Reviewer #1: Partly

Reviewer #2: Partly

2. Has the statistical analysis been performed appropriately and rigorously? 

Reviewer #1: Yes

Reviewer #2: I Don't Know

3. Have the authors made all data underlying the findings in their manuscript fully available?

Reviewer #1: Yes

Reviewer #2: No

4. Is the manuscript presented in an intelligible fashion and written in standard English?

Reviewer #1: Yes

Reviewer #2: Yes

5. Review Comments to the Author

Reviewer #1: The study found interesting and I enjoyed a lot on reviewing it. It can obviously contribute to those who are really keen on in cross-country study in WASH related themes in four Asian regions Bangledesh, Nepal, Pakistinn, and Afganistan.

I found merit in this article, however some revisions are required which are mentioned line by line in the below.

Good luck!

Abstract

1. Last sentence of background, author already spelled four South Asian countries name, thus no need to mention in this sentence. I prefer (The Multiple Indicator Cluster Survey (MICS) data collecting from the corresponding four South Asian countries serve as data source for this study.)

2. Make consistency while using abbreviation and full form whether they should be sentence case or capitalize each word such as you wrote water, sanitation, and hygiene (WASH). In the similar case you spelled out Multiple Indicator Cluster Survey (MICS). Please make consistency across the manuscript, Personally, I recommend second one.

3. Method section last line (Please verify this (Receiver Operating Characteristic (ROC) curve and area under the ROC curve (AUC).

4. The method section is not clear, it should cover methods, research design, sample, sampling procedure, and analysis strategies, including data collection date. For instance : the study applied quantitative/qualitative research method, and ……research design…. Further, it uses 3 types of analysis such as univariate, bivariate, and multivariate. For your reference you can review this article, but you are free to take an idea from any other documents.

https://journals.sagepub.com/doi/pdf/10.1177/11786302221095030

https://www.researchgate.net/publication/360175603_Effect_of_School_Water_Sanitation_and_Hygiene_on_Health_Status_Among_Basic_Level_Students'_in_Nepal

5. In the conclusion section, the sentence you have mention seems recommendation that’s fine, now please write one or two sentence as a conclusion just above the (To ensure the adequate WASH facilities, policymakers should focus on region-specific plans, providing subsidies for purchasing WASH equipment’s to the lower class of society, and conducting awareness campaigns to enhance WASH facilities in the four South Asian countries.

Title

1. Title should be "Demographic, Socioeconomic and Regional Disparities in the Coverage of Water, Sanitation, and Hygiene Facilities in Four South Asian Countries"

Introduction

1. Line 37, before writing abbreviation first write full form such as World Health Organization (WHO)/United Nations ……………(UNICEF) ..Then use only abbreviation throughout the manuscript.

2. Line 38, should be in-text citation such as ……………Ahmed et al. (2021) ………..

3. Line 41, should delete (.) before citation like …..health (United Nations, 2015).

4. Line 41 and 45, make United Nations (UN) in line 41 and use only UN in line 45 and throughout the manuscript. Apply this rule in other cases as well.

5. The introduction part is interesting, however it seems little bit lengthy thus, can you concise the introduction part only about 2 pages. Another thing is clearly state your objective of the study, how this study is different to the previous researchers, what is novelty of the study, in what ways this study findings can contribute nations and regions…..?

Methods

1. line 104 no need to write name of the Asian countries, which already mentioned in above sentence.

2. 105 and 106, no need this sentence (Figure 1 illustrates the specific datasets and their respective sample sizes used for this study.)

3. 119 use only WASH ……

4. Line 122 to 134, Regarding the (WHO/UNICEF, 2017) criteria, it is suggested to provide the JMP criteria fulfilled by selected countries for instance what was actually present in the selected countries and absent so that one can relate through findings -what facilities were suggested to be provided by the country. For example, did the improved countries had piped water as wells protected well.

The response might in this way, countries with improved WASH facilities and countries without improved WASH facilities are determined based on the guidelines provided by Joint Monitoring Programme (JMP) (2018)'s guidelines. In doing so, Bangladesh found with piped water, tube well/boreholes, protected dug wells, separate toilets for urination and defecation, ventilated improved pit latrines, fixed or portable hand-washing facilities………………………..are considered improved WASH facilities. On the other hand, countries found with unprotected dug wells, unprotected springs surface water sources, the absence of single sex toilets, ………………………the absence of hand-washing facilities, ..are considered unimproved WASH facilities.

You can separately write country wise or combined way.

5. Line 163, adjusted odds ratio (aOR),

6. Line 169-175, should be in-text citation like Ihaka and Gentleman (1996), Please follow the APA citation properly.

7. Line 192, (Table 1) Times new roman and bold, delete (:), it should be

Table 1 Country-wise household level coverage of WASH components

Follow throughout the manuscript.

8. Please mentioned the sources below the table.

9. Line 207, please follow this, Figure 3 (Italic and bold) Country-wise household level coverage of WASH facilities (times new roman)

Like Figure 3 Country-wise household level coverage of WASH facilities

10. I think, It is not AOR, should be aOR across the maniscrupt.

11. Line 273, I would love this style ……adjusted Odds Ratio (aOR)=0.54; 95% Confidence Level (CI): 0.44 to 0.65; p<0.001) and 30% lower odds (aOR=0.70; 95% CI: 0.64 to 0.77; p<0.001) instead of …………..(AOR: 0.54 with 95% CI: 0.44, 0.65) and 30% lower odds (AOR: 0.70 with 95% CI: 0.64, 0.77) respectively……

12. One important suggestion is, do not explain each and every data in the table, you may explain most significant predictors, highlighted things only and indicate please see the (table 1/2 or figure 1/2).

13. line 336 to 341 is no essential which has already mentioned introduction or in methods section, If you want to keep it please transfer in the second last paragraph of the introduction.

14. Line 346 to 357 needs triangulate this study's finding with other previous findings whether they are similar or differ. You can support by your logic the causes of having similar or contrast findings between your study's finding and others.

15. I am unable to see proper discussion, please follow these strategies while writing discussion throughout.

1. You can add one more subthemes strengths and limitation of the study or you can manage in a paragraph just above the conclusion.

Conclusion

1. Line 395 to 400 (Despite several limitations, including based on secondary data, and potential recall bias due to data collected from household member responses, our study offers a general overview of the 397 WASH conditions in the four study nations: Bangladesh, Pakistan, Nepal, and Afghanistan. There has been a minimal study on the household-level coverage of WASH facilities in Pakistan, Afghanistan, and Nepal, despite some prior research on WASH being undertaken in Bangladesh. According to our findings) seems worthless please delete these or transfer into limitations section.

2. Rewrite conclusion section, in doing so, please sketch core findings from each table and figures then only write recommendation. In the current status of your conclusion, you just mentioned recommendation. FYI, you can generate more idea from this article 10.1002/wmh3.523

Reviewer #2: Title: Demographic, Socioeconomic and Regional Disparities in the Coverage of Water, Sanitation and Hygiene (WASH) Facilities in Four South Asian Countries

Authors: Md. Hasibul Islam Jitu & Mohammad Shahed Masud

Objective: The main objective of this study was to (i)review the existing WASH facilities, (ii) mapping for regional comparisons, and (iii)identify the significant socioeconomic and demographic factors associated with WASH facilities in four South Asian countries: Bangladesh, Pakistan, Nepal, and Afghanistan.

The authors have tried to explore the situation of WASH facilities in four SAARC countries.

I have some observations on the paper and hope that this will add value to paper.

1. Abstract (Background): Add the MICS date right after the country in the bracket [such as Nepal (MICS 2019)]. Could you please present the names of countries in ascending (a to z) order, i.e., Afghanistan, Bangladesh, Nepal, and Pakistan, throughout the text? Please ensure discussion in that way. Abstract (Results): Add the value of a particular variable in the results so that the reader may understand the magnitude of the influence of variables.

2. Abstract (Conclusion): Suggest specific recommendations rather than general ones. How could policymakers use your finding into practice/intervention planning?

3. Follow the format and guidelines of PLoS One.

4. Did you get permission to use the datasets of MICS since you are going to publish them from concerned authorities?

5. In Methods: Covariates section, what is the logic for categorizing the age group <35, 35-54, and 54+, in ethnicity Bengali and others? Please present it in a logical order. For ethnicity, it may be indigenous, and others might be appropriate for all countries and categorize them accordingly. In the same way, how did you categorize poor, middle, and rich instead of five quintiles?

6. Ethical consideration: Please provide the date of ethical approval and the authority that provided the ethical clearance country-wise.

7. Please provide the date of the survey along with the country in brackets in all tables, such as Nepal (2019).

8. The authors have presented the exact description of the table. I would recommend presenting the impression of the table rather than the exact numeric value.

9. Did you assess the multi-collinearity issue before performing multivariate analysis?

10. Why and how did you claim that 'the model has quite well discriminating ability' and present it with logic and reference (citation)?

11. In the discussion session, I would recommend linking with theories related to WASH. Moreover, discuss with Sustainable WASH principles: technical, institutional, environmental, financial, and social sustainability of WASH.

12. Add the strengths and limitations of the study before the conclusion session.

13. Consult with recent and current literature.

14. Go through the text for grammar, language, and coherence.

6. PLOS authors have the option to publish the peer review history of their article (what does this mean? ). If published, this will include your full peer review and any attached files.

**Do you want your identity to be public for this peer review?** For information about this choice, including consent withdrawal, please see our Privacy Policy .

Reviewer #1: **Yes: ** Mohan Kumar Sharma, PhD.

Reviewer #2: No

---

## [Author Response · Author response to Decision Letter 1]

9 Oct 2024

RESPONSES TO EDITOR ON THE JOURNAL REQUIREMENTS

Requirement: Please provide a link to the terms of use of the R package used as some of the R packages source their maps from Google maps which is not compliant with our licensing rules. Please provide the link to its sources so we may confirm it is not Google maps.

Authors’ response: Apologies for any confusion. Actually, we did not use any R packages that source their maps from Google Maps. Instead, we utilized the administrative boundary shapefile of South Asian countries from an openly licensed database, the geoBoundaries Global Administrative Database (Runfola et al., 2020), which we have cited in our updated manuscript. The R packages were only used to manage the shapefiles and specifically for plotting them. We have cited the main packages used for our analysis. The terms of use are available on CRAN website. Additionally, we have uploaded the R code and corresponding shapefiles as a zip file titled "R_codes_for_WASH_mapping_(Not for publishing)_PONE-D-24-27103R1" to validate our claims. However, we do not want to publish the R code; it is provided to check the journal's requirements about mapping.

Packages the terms of use

sf https://cran.r-project.org/web/packages/sf/index.html

tidyverse https://cran.r-project.org/web/packages/tidyverse/index.html

ggplot2 https://cran.r-project.org/web/packages/ggplot2/index.html

gridExtra https://cran.r-project.org/web/packages/gridExtra/index.html

Reference: https://www.geoboundaries.org/countryDownloads.html

Runfola D, Anderson A, Baier H, Crittenden M, Dowker E, Fuhrig S, Goodman S, Grimsley G, Layko R, Melville G, Mulder M. geoBoundaries: A global database of political administrative boundaries. PloS one. 2020 Apr 24;15(4):e0231866.

Requirement: Please ensure that your manuscript meets PLOS ONE's style requirements, including those for file naming. The PLOS ONE style templates can be found at

Authors’ response: Checked and its ok.

Requirement: We note that Figures 3 and 4 in your submission contain [map/satellite] images which may be copyrighted. All PLOS content is published under the Creative Commons Attribution License (CC BY 4.0), which means that the manuscript, images, and Supporting Information files will be freely available online, and any third party is permitted to access, download, copy, distribute, and use these materials in any way, even commercially, with proper attribution. For these reasons, we cannot publish previously copyrighted maps or satellite images created using proprietary data, such as Google software (Google Maps, Street View, and Earth). For more information, see our copyright guidelines: http://journals.plos.org/plosone/s/licenses-and-copyright.

Authors’ response: Figures 3 and 4 in our manuscript contain spatial mappings of the study countries and regions. These maps are not copyrighted. They are original images created by us using the statistical software R. The R packages and shapefile used in this manuscript are listed on page 8; in the last paragraph of the Statistical analyses section.

RESPONSES TO THE COMMENTS OF REVIEWER #1

Reviewer’s comment: Last sentence of background, author already spelled four South Asian countries name, thus no need to mention in this sentence. I prefer (The Multiple Indicator Cluster Survey (MICS) data collecting from the corresponding four South Asian countries serve as data source for this study.)

Authors’ response: Background is revised and the comment is corrected.

Reviewer’s comment: Make consistency while using abbreviation and full form whether they should be sentence case or capitalize each word such as you wrote water, sanitation, and hygiene (WASH). In the similar case you spelled out Multiple Indicator Cluster Survey (MICS). Please make consistency across the manuscript, Personally, I recommend second one.

Authors’ response: Corrected.

Reviewer’s comment: Method section last line (Please verify this (Receiver Operating Characteristic (ROC) curve and area under the ROC curve (AUC).

Authors’ response: Corrected. Thank you for pointing this out. Since AUC values already represent the area under the ROC curve, it does not need to be mentioned again in the methods section.

Reviewer’s comment: The method section is not clear, it should cover methods, research design, sample, sampling procedure, and analysis strategies, including data collection date. For instance: the study applied quantitative/qualitative research method, and ……research design…. Further, it uses 3 types of analysis such as univariate, bivariate, and multivariate. For your reference you can review this article, but you are free to take an idea from any other documents. https://journals.sagepub.com/doi/pdf/10.1177/11786302221095030

https://www.researchgate.net/publication/360175603_Effect_of_School_Water_Sanitation_and_Hygiene_on_Health_Status_Among_Basic_Level_Students'_in_Nepal

Authors’ response: The method section has been revised based on the ideas from the paper you suggested.

Reviewer’s comment: In the conclusion section, the sentence you have mention seems recommendation that’s fine, now please write one or two sentence as a conclusion just above the (To ensure the adequate WASH facilities, policymakers should focus on region-specific plans, providing subsidies for purchasing WASH equipment’s to the lower class of society, and conducting awareness campaigns to enhance WASH facilities in the four South Asian countries.

Authors’ response: We have revised the conclusion.

Reviewer’s comment: Title should be "Demographic, Socioeconomic and Regional Disparities in the Coverage of Water, Sanitation, and Hygiene Facilities in Four South Asian Countries"

Authors’ response: Corrected.

Reviewer’s comment: Line 37, before writing abbreviation first write full form such as World Health Organization (WHO)/United Nations ……………(UNICEF) ..Then use only abbreviation throughout the manuscript.

Authors’ response: Corrected.

Reviewer’s comment: Line 38, should be in-text citation such as ……………Ahmed et al. (2021) ………..

Authors’ response: Corrected.

Reviewer’s comment: Line 41, should delete (.) before citation like …..health (United Nations, 2015).

Authors’ response: Corrected.

Reviewer’s comment: The introduction part is interesting, however it seems little bit lengthy thus, can you concise the introduction part only about 2 pages. Another thing is clearly state your objective of the study, how this study is different to the previous researchers, what is novelty of the study, in what ways this study findings can contribute nations and regions…..?

Authors’ response: The introduction has been condensed to approximately two pages. Now, it clearly outlines the objectives, highlights the differences from previous studies, and emphasizes the novelty of the research, along with its contributions to both national and regional contexts.

Reviewer’s comment: line 104 no need to write name of the Asian countries, which already mentioned in above sentence.

Authors’ response: Corrected.

Reviewer’s comment: 105 and 106, no need this sentence (Figure 1 illustrates the specific datasets and their respective sample sizes used for this study.)

Authors’ response: Removed.

Reviewer’s comment: 119 use only WASH ……

Authors’ response: Corrected.

Reviewer’s comment: Line 122 to 134, Regarding the (WHO/UNICEF, 2017) criteria, it is suggested to provide the JMP criteria fulfilled by selected countries for instance what was actually present in the selected countries and absent so that one can relate through findings -what facilities were suggested to be provided by the country. For example, did the improved countries had piped water as wells protected well. The response might in this way, countries with improved WASH facilities and countries without improved WASH facilities are determined based on the guidelines provided by Joint Monitoring Programme (JMP) (2018)'s guidelines. In doing so, Bangladesh found with piped water, tube well/boreholes, protected dug wells, separate toilets for urination and defecation, ventilated improved pit latrines, fixed or portable hand-washing facilities………………………..are considered improved WASH facilities. On the other hand, countries found with unprotected dug wells, unprotected springs surface water sources, the absence of single sex toilets, ………………………the absence of hand-washing facilities, ..are considered unimproved WASH facilities. You can separately write country wise or combined way.

Authors’ response: Corrected. All four countries used the same criteria for levelling the corresponding three outcome variables: Water, Sanitation, and Hygiene. In the MICS summary reports, the JMP guidelines were also used to categorize facilities as improved or unimproved. We verified this from each of the MICS summary reports of the corresponding surveys available online. Additionally, a previous study in Bangladesh used the same approach: https://doi.org/10.1371/journal.pone.0259635

Reviewer’s comment: Line 163, adjusted odds ratio (aOR),

Authors’ response: Corrected.

Reviewer’s comment: Line 169-175, should be in-text citation like Ihaka and Gentleman (1996), Please follow the APA citation properly.

Authors’ response: Corrected.

Reviewer’s comment: Line 192, (Table 1) Times new roman and bold, delete (:), it should be

Table 1 Country-wise household level coverage of WASH components

Follow throughout the manuscript.

Authors’ response: Corrected.

Reviewer’s comment: Please mentioned the sources below the table.

Authors’ response: All the values in the table are the results of our analysis and were not taken from any other sources. Therefore, sources are not mentioned below the table.

Reviewer’s comment: Line 207, please follow this, Figure 3 (Italic and bold) Country-wise household level coverage of WASH facilities (times new roman)

Like Figure 3 Country-wise household level coverage of WASH facilities

Authors’ response: Corrected.

Reviewer’s comment: I think, It is not AOR, should be aOR across the maniscrupt.

Authors’ response: Corrected.

Reviewer’s comment: Line 273, I would love this style ……adjusted Odds Ratio (aOR)=0.54; 95% Confidence Level (CI): 0.44 to 0.65; p<0.001) and 30% lower odds (aOR=0.70; 95% CI: 0.64 to 0.77; p<0.001) instead of …………..(AOR: 0.54 with 95% CI: 0.44, 0.65) and 30% lower odds (AOR: 0.70 with 95% CI: 0.64, 0.77) respectively……

Authors’ response: Revised.

Reviewer’s comment: One important suggestion is, do not explain each and every data in the table, you may explain most significant predictors, highlighted things only and indicate please see the (table 1/2 or figure 1/2).

Authors’ response: Revised.

Reviewer’s comment: line 336 to 341 is no essential which has already mentioned introduction or in methods section, If you want to keep it please transfer in the second last paragraph of the introduction.

Authors’ response: These lines have been moved to the introduction and rewritten.

Reviewer’s comment: Line 346 to 357 needs triangulate this study's finding with other previous findings whether they are similar or differ. You can support by your logic the causes of having similar or contrast findings between your study's finding and others.

Authors’ response: Corrected.

Reviewer’s comment: I am unable to see proper discussion, please follow these strategies while writing discussion throughout.

Authors’ response: We revised the whole discussion section.

Reviewer’s comment: You can add one more subthemes strengths and limitation of the study or you can manage in a paragraph just above the conclusion.

Authors’ response: Added.

Reviewer’s comment: Line 395 to 400 (Despite several limitations, including based on secondary data, and potential recall bias due to data collected from household member responses, our study offers a general overview of the WASH conditions in the four study nations: Bangladesh, Pakistan, Nepal, and Afghanistan. There has been a minimal study on the household-level coverage of WASH facilities in Pakistan, Afghanistan, and Nepal, despite some prior research on WASH being undertaken in Bangladesh. According to our findings) seems worthless please delete these or transfer into limitations section.

Authors’ response: Corrected.

Reviewer’s comment: Rewrite conclusion section, in doing so, please sketch core findings from each table and figures then only write recommendation. In the current status of your conclusion, you just mentioned recommendation. FYI, you can generate more idea from this article 10.1002/wmh3.523

Authors’ response: Revised.

RESPONSES TO THE COMMENTS OF REVIEWER #2

Reviewer’s comment: 1. Abstract (Background): Add the MICS date right after the country in the bracket [such as Nepal (MICS 2019)]. Could you please present the names of countries in ascending (a to z) order, i.e., Afghanistan, Bangladesh, Nepal, and Pakistan, throughout the text? Please ensure discussion in that way. Abstract (Results): Add the value of a particular variable in the results so that the reader may understand the magnitude of the influence of variables.

Authors’ Response: We have addressed all the points you raised. In the Abstract (Background), we added the MICS data and corresponding year and moved this information to the Methods section, as the data source was originally a subsection of the methodology in our manuscript. In the Abstract (Results), we have now included the relevant values.

Reviewer’s comment: 2. Abstract (Conclusion): Suggest specific recommendations rather than general ones. How could policymakers use your finding into practice/intervention planning?

Authors’ Response: We have revised this conclusion using specific recommendations.

Reviewer’s comment: 3. Follow the format and guidelines of PLoS One.

Authors’ Response: Followed.

Reviewer’s comment: 4. Did you get permission to use the datasets of MICS since you are going to publish them from concerned authorities?

Authors’ Response: Yes, we applied for the specific datasets required for our study from the MICS team, and they approved our access to the datasets via email.

Reviewer’s comment: 5. In Methods: Covariates section, what is the logic for categorizing the age group <35, 35-54, and 54+, in ethnicity Bengali and others? Please present it in a logical order. For ethnicity, it may be indigenous, and others might be appropriate for all countries and categorize them accordingly. In the same way, how did you categorize poor, middle, and rich instead of five quintiles?

Authors’ Response: The age groups were categorized as <35, 35-54, and 54+ following the methodology in Armah et al. (2018).

Reference: Armah, F. A., Ekumah, B., Yawson, D. O., Odoi, J. O., Afitiri, A. R., & Nyieku, F. E. (2018). Access to improved water and sanitation in sub-Saharan Africa in a quarter century. Heliyon, 4(11). https://doi.org/10.1016/j.heliyon.2018.e00931.

For Bangladesh, the MICS 2019 data provides only two categories for ethnicity: Bengali and Others. The other countries did not include ethnicity information, which is why we were unable to consider it. However, the previous studies had found ethnic disparities in access to WASH in Bangladesh (Alam, 2022).

Reference: Alam MZ. Ethnic inequalities in access to WASH in Bangladesh. The Lancet Global Health. 2022 Aug 1;10(8):e1086-7. https://doi.org/10.1016/S2214-109X(22)00232-7

The five quintiles were grouped into three categories as follows: 'Poorest' and 'Second/Poorer' were combined into 'Poor'; 'Middle' remained as 'Middle'; and 'Fourth/Richer' and 'Richest' were combined into 'Rich'. This categorization was adopted from Dhital et al. (2022).

Reference: Dhital, S. R., Chojenta, C., Evans, T.-J., Acharya, T. D., & Loxton, D. (2022). Prevalence and correlates of Water, Sanitation, and Hygiene (WASH) and spatial distribution of unimproved WASH in Nepal. International Journal of Environmental Research and Public Health, 19(3507). https://doi.org/10.3390/ijerph19063507.

Reviewer’s comment: 6. Ethical consideration: Please provide the date of ethical approval and the authori

---

## [Decision Letter · Decision Letter 1]

7 Feb 2025

Demographic, Socioeconomic and Regional Disparities in the Coverage of Water, Sanitation and Hygiene Facilities in Four South Asian Countries

PONE-D-24-27103R1

Dear Dr. Md. Hasibul Islam Jitu,

We’re pleased to inform you that your manuscript has been judged scientifically suitable for publication and will be formally accepted for publication once it meets all outstanding technical requirements.

Kind regards,

Ramesh Adhikari

Academic Editor

PLOS ONE

Additional Editor Comments (optional):

Reviewers' comments:

Reviewer's Responses to Questions

**Comments to the Author**

1. If the authors have adequately addressed your comments raised in a previous round of review and you feel that this manuscript is now acceptable for publication, you may indicate that here to bypass the “Comments to the Author” section, enter your conflict of interest statement in the “Confidential to Editor” section, and submit your "Accept" recommendation.

Reviewer #1: All comments have been addressed

Reviewer #2: All comments have been addressed

2. Is the manuscript technically sound, and do the data support the conclusions?

Reviewer #1: Yes

Reviewer #2: Yes

3. Has the statistical analysis been performed appropriately and rigorously? 

Reviewer #1: Yes

Reviewer #2: Yes

4. Have the authors made all data underlying the findings in their manuscript fully available?

Reviewer #1: Yes

Reviewer #2: Yes

5. Is the manuscript presented in an intelligible fashion and written in standard English?

Reviewer #1: Yes

Reviewer #2: Yes

6. Review Comments to the Author

Reviewer #1: I appreciate such a academic task, authors have addressed all of my concerns and seems good to proceed. However, authors, have you missed speall keywords in this revised version.

Thank you and good luck.

Reviewer #2: (No Response)

7. PLOS authors have the option to publish the peer review history of their article (what does this mean? ). If published, this will include your full peer review and any attached files.

**Do you want your identity to be public for this peer review?** For information about this choice, including consent withdrawal, please see our Privacy Policy .

Reviewer #1: No

Reviewer #2: **Yes: ** Devaraj Acharya

---

## [Editor Report · Acceptance letter]

PONE-D-24-27103R1

PLOS ONE

Dear Dr. Jitu,

I'm pleased to inform you that your manuscript has been deemed suitable for publication in PLOS ONE. Congratulations! Your manuscript is now being handed over to our production team.

Kind regards,

on behalf of

Professor Ramesh Adhikari

Academic Editor

PLOS ONE